# The possible molecular mechanism underlying the involvement of the variable shear factor QKI in the epithelial-mesenchymal transformation of oesophageal cancer

Yishuang Cui[1,2], Yanan Wu[1,2], Yingze Zhu[2,3,4], Wei Liu[2,3,4], Lanxiang Huang[2,3,4], Ziqian Hong[1,2], Mengshi Zhang[1,2], Xuan Zheng[1,2], Guogui Sun[1,2,3,4]*

1 School of Public Health, North China University of Science and Technology, Tangshan, Hebei Province, China, 2 Department of Hebei Key Laboratory of Medical-Industrial Integration Precision Medicine, Tangshan, Hebei Province, China, 3 School of Clinical Medicine, North China University of Science and Technology, Tangshan, Hebei Province, China, 4 Affiliated Hospital, North China University of Science and Technology, Tangshan, Hebei Province, China

* guogui_sun2021@sina.com

**Data Availability Statement:** All relevant data are within the paper and its Supporting Information files.

## Abstract

### Objective

Based on the GEO, TCGA and GTEx databases, we reveal the possible molecular mechanism of the variable shear factor QKI in epithelial mesenchymal transformation (EMT) of oesophageal cancer.

### Methods

Based on the TCGA and GTEx databases, the differential expression of the variable shear factor QKI in oesophageal cancer samples was analysed, and functional enrichment analysis of QKI was performed based on the TCGA-ESCA dataset. The percent-spliced in (PSI) data of oesophageal cancer samples were downloaded from the TCGASpliceSeq database, and the genes and variable splicing types that were significantly related to the expression of the variable splicing factor QKI were screened out. We further identified the significantly upregulated circRNAs and their corresponding coding genes in oesophageal cancer, screened the EMT-related genes that were significantly positively correlated with QKI expression, predicted the circRNA-miRNA binding relationship through the circBank database, predicted the miRNA-mRNA binding relationship through the TargetScan database, and finally obtained the circRNA-miRNA-mRNA network through which QKI promoted the EMT process.

### Results

Compared with normal control tissue, QKI expression was significantly upregulated in tumour tissue samples of oesophageal cancer patients. High expression of QKI may

**Funding:** General Project of National Natural Science Foundation of China (82172658); Tangshan small cell lung cancer medical-industrial integration precision diagnosis and treatment basic innovation team (21130203D). The funders had no role in study design, data collection and analysis, decision to publish, or preparation of the manuscript.

**Competing interests:** The authors have declared that no competing interests exist.

promote the EMT process in oesophageal cancer. QKI promotes hsa_circ_0006646 and hsa_circ_0061395 generation by regulating the variable shear of BACH1 and PTK2. In oesophageal cancer, QKI may promote the production of the above two circRNAs by regulating variable splicing, and these circRNAs further competitively bind miRNAs to relieve the targeted inhibition of IL-11, MFAP2, MMP10, and MMP1 and finally promote the EMT process.

## Conclusion

Variable shear factor QKI promotes hsa_circ_0006646 and hsa_circ_0061395 generation, and downstream related miRNAs can relieve the targeted inhibition of EMT-related genes (IL11, MFAP2, MMP10, MMP1) and promote the occurrence and development of oesophageal cancer, providing a new theoretical basis for screening prognostic markers of oesophageal cancer patients.

## Introduction

In 2020, there will be 604,000 new cases of oesophageal cancer and more than 540,000 deaths worldwide [1]. Oesophageal cancer ranks sixth in the global cancer mortality rate. East Asia has the highest incidence rate, of which China has the highest incidence of oesophageal squamous cell cancer [2–4]. Research shows that most patients with oesophageal cancer are in the advanced stage of cancer at the time of treatment, and their prognosis is poor, and the recurrence rate is high [5–8]. Most importantly, the response of oesophageal cancer patients to existing chemotherapy and radiotherapy methods is highly heterogeneous [9, 10]. Therefore, more biomarkers need to be explored to treat patients with oesophageal cancer.

Circular RNAs (circRNAs) have a unique covalent closed loop structure, so their stability is excellent, and their expression is relatively conserved. CircRNAs can participate in a variety of biological processes, among which circRNAs can regulate the expression of their downstream target genes through interactions with RNA-binding proteins [11–13]. The cyclic RNA-binding protein QKI is located on human chromosome 6. QKI specifically binds RNA through tremor response elements [14–17]. QKI also participates in epithelial-mesenchymal transformation (EMT) by splicing and regulating the formation of circRNAs. During EMT, siRNAs targeting approximately 20 RBPs were used. Changes in circRNA levels were observed only when the expression of QKI protein decreased, indicating that QKI was needed to generate circRNAs [18–20]. Hundreds of circRNAs are regulated during human EMT, and more than one-third of the production of rich cyclic RNA is dynamically regulated by the variable splicing factor QKI, which promotes the formation of cyclic RNA by combining with the typical motif (ACUAACN-20UAAC motif) on the lateral intron of cyclic RNA [21]. For example, QKI can promote the production of circRNAs in the EMT process of breast epithelial cells [22].

It has been reported that the RNA binding protein QKI is related to the occurrence and development of various cancers, but its role in oesophageal cancer is still poorly understood. EMT is a necessary cellular process for the occurrence and metastasis of various tumours [23–25]. It is important to analyse the molecular mechanism by which QKI regulates circRNAs participating in EMT in oesophageal cancer for the prevention and treatment of oesophageal cancer.

## Methods

### Data download from gene expression omnibus (GEO), The cancer genome atlas (TCGA), and genotype-tissue expression (GTEx) databases

The TCGA databases (https://www.cancer.gov) was used to download the gene expression data of normal control and oesophageal cancer patients, and the downloaded data format was transcripts per million (TPM) reads. Since the number of normal control samples in the TCGA database was small, we selected the GTEx database (https://www.GTEXportal.org/) to download normal oesophageal tissue samples from healthy humans. A total of 286 normal control tissue samples and 182 tumour tissue samples from oesophageal cancer patients were collected.

The GEO database (http://www.ncbi.nlm.nih.gov/geo/) was used to retrieve the oesophageal carcinoma-related transcriptome sequencing dataset GSE189830 and download the circRNA expression profile (normal group, n = 3; tumour group, n = 3). Since these data were obtained from public databases, no ethical approval or informed consent was needed.

### Screening of differentially expressed mRNAs and circRNAs

The ComBat R package was used to remove the batch effect of data collected from TCGA and GTEx databases, after which data were merged. Then the limma R package was applied to standardize the gene expression data, and analyze the differential expression of QKI in normal control tissue samples and tumour tissue samples from oesophageal cancer patients. The nonparametric Wilcoxon rank sum test was used when the normality assumption was not satisfied. Similarly, the limma package was used to perform differential analysis on the GSE189830 dataset to obtain differentially expressed circRNAs. All the analyses in this paper were conducted under R version 4.2.1 (R Foundation for Statistical Computing).

### Gene set enrichment analysis (GSEA)

Gene set enrichment analysis (GSEA) is a computational method for determining whether a priori-defined gene set shows statistically significant consistent differences between two biological states. In this study, we analyzed the TCGA-ESCA dataset based on the QKI median expression value to analyse the high and low expression groups of QKI. We compared the enriched pathway differences between the two groups by referring to the HALLMARK dataset. Then, we selected the path with the most significant difference with the normalized enrichment score (NES) $p$ value and false discovery rate (FDR) $q$ value as the criteria and drew the enrichment plot. Larger |NES| values of each gene set indicate that the gene set exhibits a stronger enrichment signal between the high and low expression groups of QKI. If |NES| values are more than 0, the gene set shows higher activity in the high expression group of QKI; if less than 0, the gene set is more active in the low expression group of QKI.

### Download of alternative splicing data from the TCGA SpliceSeq database

Through the TCGASpliceSeq database (https://bioinformatics.mdanderson.org/TCGASpliceSeq/), the percent spliced in (PSI) data of oesophageal cancer samples were downloaded. alternative splicing modes were analyzed, and an UpSet diagram of alternative splicing (schematic diagram of different alternative splicing methods) was drawn.

PSI can be calculated for isoform, exon and atomic simulation environment (ASE). For ASE, PSI = splice_ In/ (splice in + splice out), splice_ in and splice_ out in RNA-seq data is the number of reads that support splice in and splice out.

### Obtaining EMT-related genes from the MSigDB database

In the MSigDB database (https://www.gsea-msigdb.org/gsea/index.jsp) we searched for epithelial-mesenchymal transformation and downloaded FOROUTAN_INTEGRATED_TGF-B_EMT_ UP.v2022.1.Hs.gmt.

### Correlation analysis

Based on the variable splicing data downloaded from the TCGASpliceSeq database, the correlation between QKI and gene alternative splicing was analysed through the R software package coreplot.

Based on the tumour tissue samples of oesophageal carcinoma patients in the TCGA-ESCA dataset, the correlation between QKI and the expression of IL-11, MFAP2, MMP10 and MMP1 in oesophageal carcinoma samples was analysed by using the R software package coreplot, the correlation coefficient was calculated, and a correlation scatter diagram was drawn.

### Prediction of the circRNA-miRNA-mRNA binding relationship

The circBank database (http://www.circbank.cn/) was used to predict the relationship between circRNAs and miRNAs. The TargetScan database (https://www.targetscan.org/vert_71/) was used to predict the relationship between miRNA and mRNA and screen with context++-score<-0.20 as the threshold.

### Construction of the circRNA-miRNA-mRNA regulatory network

Through Cytoscape software (v3.8.0), the network diagram depicting the relationship between QKI and its regulatory gene alternative splicing mode was drawn. In addition, according to the circRNA-miRNA-mRNA relationship, the circRNA-miRNA-mRNA regulatory network diagram was drawn using Cytoscape software. Finally, the Xiantao Academic Online Tool (https://www.xiantao.love/) was used to draw the Sangi diagram of the QKI-circRNA-miRNA-mRNA-EMT regulatory network.

## Results

### Variable shear factor QKI is closely related to epithelial-mesenchymal transformation (EMT) in oesophageal cancer

We combined the data collected from the TCGA-ESCA and GTEx datasets. After data correction, an analysis of differential expression showed that the expression of QKI in tumour tissue samples of oesophageal carcinoma patients was significantly upregulated compared with those in normal control samples (Fig 1A). The tumour tissue samples of oesophageal carcinoma patients in the TCGA-ESCA dataset were divided into a high expression QKI group and a low expression QKI group. GSEA found that the EMT-related pathway ranked first (NES = 3.31, $p$ value < 0.0001, FDR $q$ value < 0.0001), which showed that high expression of QKI could promote the EMT process (Fig 1B). The above results preliminarily indicate that QKI may promote the EMT process in oesophageal carcinoma.

### Variable shear factor QKI can promote hsa_ circ_ 0006646 and hsa_ circ_ 0061395 generation

To further explore the downstream coding gene regulated by the variable shear factor QKI in oesophageal cancer, we downloaded the PSI data of oesophageal cancer samples through the TCGASpliceSeq database. The different variable shear patterns were analysed, and the

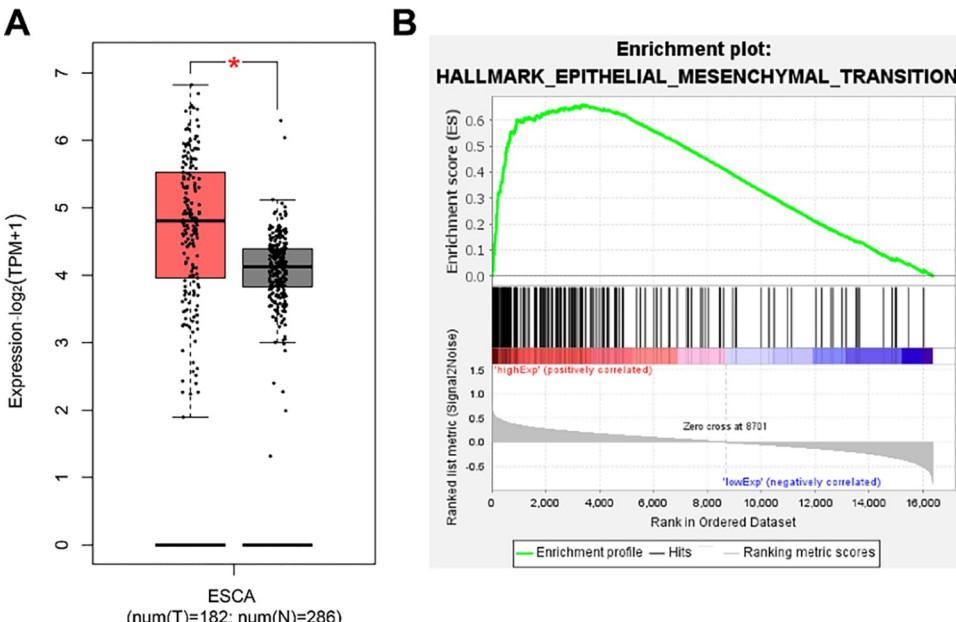

**Fig 1. Analysis of differential expression of QKI and pathway enrichment.** Expression of QKI in normal samples and tumour tissue samples from patients with oesophageal carcinoma (normal group, n = 286; tumour group, n = 182) (A). * p.adj < 0.05. GSEA showed the enriched pathway differences between the high and low expression groups of QKI; ESCA: Oesophageal carcinoma (B).

alternative splicing UpSet diagram was drawn. As shown in Fig 2A, there was a large amount of alternative splicing in oesophageal carcinoma. The height of the columns in the figure indicates the number of genes that undergo certain alternative splicing. The first three columns are the individual parts of exon skip (ES), alternate terminator (AT), and alternate promoter (AP), the fourth column indicates the common part of ES and AP, and the fifth column indicates the common part of ES, AT, and AP, which fully showed that the proportion of ES, AT, and AP in oesophageal carcinoma was relatively high. Then, we screened out the genes (50) and their variable shear types (52) significantly related to the expression of the variable shear factor QKI by correlation analysis, taking | r |>0.2 and *p*. adjusted <0.05 as the threshold (Fig 2B; Table 1). At the same time, we screened the significantly upregulated circRNAs and their corresponding coding genes in 72 oesophageal carcinomas through the GEO dataset GSE189830 (Fig 2C). Fifty coding genes related to QKI and 72 genes encoding differentially expressed circRNAs were intersected to obtain 2 overlapping genes, namely, BACH1 and PTK2 (Fig 2D). Both BACH1 and PTK2 have AP modes of variable shearing and are encoded as hsa_circ_0006646 and hsa_circ_0061395, respectively. Therefore, we believe that in oesophageal cancer, the variable shear factor QKI can promote hsa_circ_0006646 and hsa_circ_0061395 generation.

## QKI expression is significantly positively correlated with the expression of EMT-related genes (IL11, MFAP2, MMP10, MMP1)

Based on the TCGA-ESCA dataset, we further screened the genes that were significantly positively correlated with QKI expression. First, with |logFC|>3 and p.adj < 0.05 as the threshold, 204 genes significantly upregulated in oesophageal cancer were screened (Fig 3A). Through the MSigDB database, 120 genes related to EMT activation were retrieved and further intersected with more than 204 genes to obtain four intersecting genes, namely, IL11, MFAP2, MMP10, and MMP1 (Fig 3B). These four intersecting genes may play an important role in the

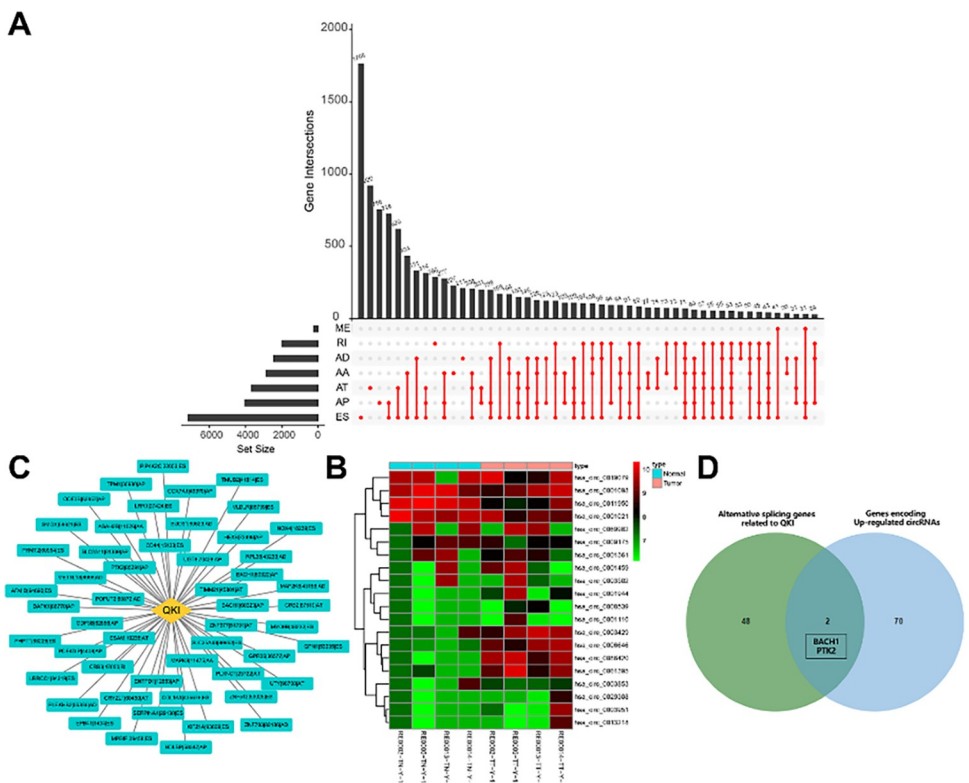

**Fig 2. Screening of circRNAs downstream of the variable shear factor QKI.** Seven kinds of variable clipping UpSet diagrams (A). The network relationship between QKI and its regulatory gene variable splicing pattern (B). Heatmap for differentially expressed genes in the GSE189830 dataset (normal group, n = 3; tumour group, n = 3). (C). The expression values are represented in different colors; red indicates high expression, and green indicates low expression. Darker node colors indicate more extreme high or low expression of the respective gene. Venn diagram of the intersection of 50 coding genes related to QKI and 72 genes encoding differentially expressed circRNAs (D).

EMT of oesophageal cancer. The results of correlation analysis showed that QKI was significantly positively correlated with the expression of IL-11, MFAP2, MMP10 and MMP1 (Fig 3C–3F). This suggests that QKI may activate the expression of IL-11, MFAP2, MMP10, and MMP1, thus promoting the EMT process.

## QKI promotes EMT by regulating the circRNA-miRNA-mRNA network

We predicted the downstream miRNAs of hsa_circ_0006646 and hsa_circ_0061395 through the circBank database and predicted the upstream miRNAs of IL11, MFAP2, MMP10, and MMP1 through the TargetScan database. The above two were intersected to obtain the circRNA-miRNA-mRNA regulatory network. We found that hsa_circ_0006646 may bind hsa-miR-4708-3p, hsa-miR-4778-3p, hsa-miR-4778-5p, hsa-miR-3180-5p, hsa-miR-425-5p, hsa-miR-5589-3p, hsa-miR-6888-3p, hsa-miR-4743-3p, hsa-miR-488-3p, hsa-miR-7161-3p, hsa-miR-130a-5p, hsa-miR-4712-5p, hsa-miR-770-5p, hsa-miR-548ae-3p, hsa-miR-548ah-3p, hsa-miR-548ah-3p, has-miR-548aq-3p, hsa-miR-548j-3p, hsa-miR-7155-5p, and then release the targeted inhibition of IL11, MFAP2, MMP10 and MMP1 (Fig 4A; Tables 2 and 4). Furthermore, hsa_circ_0061395 may relieve the targeted inhibition of IL-11, MFAP2, MMP10 and MMP1 by competitively binding hsa-miR-369-3p, hsa-miR-3118, hsa-miR-342-3p, hsa-miR-4712-5p, hsa-miR-770-5p and hsa-miR-6865-5p (Fig 4B; Tables 3 and 4). Thus far, we reached the following conclusion: in oesophageal cancer, the variable shear factor QKI can promote

**Table 1. Correlation analysis results of variable splicing of QKI and its downstream coding genes.**

| Splicing factor | Gene name | AS ID | AS type | r | *p*. adjusted | correlation |
|---|---|---|---|---|---|---|
| QKI | RPL38 | 43233 | AD | 0.226 | 0.009 | positive |
| | SLC2A11 | 61339 | AP | -0.231 | 0.008 | negative |
| | PDE4DIP | 4408 | AP | -0.225 | 0.009 | negative |
| | HEXB | 72498 | AP | -0.296 | 0.002 | negative |
| | COL1A1 | 435679 | ES | 0.205 | 0.012 | positive |
| | SERPINA1 | 29130 | ES | -0.319 | 0.001 | negative |
| | ODF3B | 62857 | AP | -0.222 | 0.009 | negative |
| | ODF3B | 62856 | AP | 0.222 | 0.009 | positive |
| | POFUT2 | 60872 | AD | -0.235 | 0.008 | negative |
| | PTK2 | 85291 | AP | 0.295 | 0.002 | positive |
| | CRB3 | 47050 | RI | -0.214 | 0.011 | negative |
| | NOX4 | 18239 | ES | -0.236 | 0.007 | negative |
| | PLXNC1 | 23722 | AT | 0.201 | 0.012 | positive |
| | AFMID | 94690 | ES | -0.215 | 0.010 | negative |
| | ZNF783 | 82186 | AD | 0.208 | 0.011 | positive |
| | UTY | 90708 | AT | 0.265 | 0.004 | positive |
| | PRMT2 | 60964 | ES | 0.267 | 0.004 | positive |
| | CD44 | 15133 | ES | -0.245 | 0.006 | negative |
| | BUD31 | 80623 | AD | 0.264 | 0.004 | positive |
| | ZNF677 | 51701 | AT | -0.236 | 0.007 | negative |
| | METTL13 | 8998 | AD | 0.271 | 0.004 | positive |
| | COX7A1 | 49370 | AP | -0.233 | 0.008 | negative |
| | VLDLR | 85739 | ES | 0.230 | 0.008 | positive |
| | PHPT1 | 88225 | ES | -0.266 | 0.004 | negative |
| | ASAH2B | 11576 | AA | 0.206 | 0.012 | positive |
| | PIP4K2C | 22653 | ES | 0.211 | 0.011 | positive |
| | GPR56 | 36577 | AP | -0.344 | 0.000 | negative |
| | SLC25A43 | 89952 | ES | -0.228 | 0.008 | negative |
| | UGT8 | 70428 | AP | 0.265 | 0.004 | positive |
| | PLEKHB2 | 55376 | AD | 0.215 | 0.010 | positive |
| | MAP2K6 | 43188 | AD | 0.200 | 0.013 | positive |
| | ESAM | 19235 | AT | 0.339 | 0.000 | positive |
| | MAPK8 | 11475 | AA | 0.221 | 0.009 | positive |
| | ZNF512 | 53026 | ES | -0.203 | 0.012 | negative |
| | TMUB2 | 41814 | ES | -0.252 | 0.005 | negative |
| | HDLBP | 58347 | AP | -0.272 | 0.004 | negative |
| | LRR1 | 27424 | ES | 0.270 | 0.004 | positive |
| | GPN1 | 53035 | ES | -0.206 | 0.012 | negative |
| | EPB41 | 1404 | ES | 0.200 | 0.013 | positive |
| | CRB2 | 87510 | AT | -0.209 | 0.011 | negative |
| | TIMM21 | 45801 | AT | 0.246 | 0.006 | positive |
| | DAPK1 | 86770 | AP | 0.204 | 0.012 | positive |
| | CRYZL1 | 60450 | AT | -0.249 | 0.006 | negative |
| | MYO9B | 48232 | ES | 0.204 | 0.012 | positive |
| | ENTPD1 | 12653 | AP | 0.212 | 0.011 | positive |
| | KIF21A | 93029 | ES | -0.300 | 0.002 | negative |
| | MPRIP | 39458 | ES | -0.273 | 0.004 | negative |

(*Continued*)

**Table 1.** (Continued)

| Splicing factor | Gene name | AS ID | AS type | r | p. adjusted | correlation |
|---|---|---|---|---|---|---|
| | TPM1 | 30980 | AP | -0.423 | 0.000 | negative |
| | SMOX | 58621 | ES | -0.269 | 0.004 | negative |
| | BACH1 | 60323 | AP | -0.212 | 0.011 | negative |
| | BACH1 | 60322 | AP | 0.212 | 0.011 | positive |
| | LRRCC1 | 84319 | ES | -0.234 | 0.008 | negative |

hsa_ circBACH1_ 012, and hsa_circ_0061395 generation, and these circRNAs further competitively bind miRNAs to remove the targeted inhibition of IL11, MFAP2, MMP10 and MMP1 and finally promote the EMT process (Fig 4C).

## Discussion

At present, there are few molecular mechanisms known to be involved in the occurrence, development and metastasis of oesophageal cancer. The commonly used prediction and treatment methods are not specific and sensitive for individuals. Therefore, more biomarkers need to be explored for the early diagnosis of oesophageal cancer patients. Based on the stability of circRNA expression and other characteristics, in-depth study of the molecular function and mechanism of circRNA in oesophageal cancer will be very beneficial to the clinical diagnosis and prognosis of oesophageal cancer. Research shows that circRNAs participate in the pathological process of many diseases, including oesophageal cancer, and can be used as new targets for cancer treatment [26, 27]. A systematic review and meta-analysis showed that the analysis of plasma circRNA expression was a highly sensitive manner of identifying ESCC patients [28]. For example, hsa_circ_0062459 and hsa_circ_0043603 have potential as blood

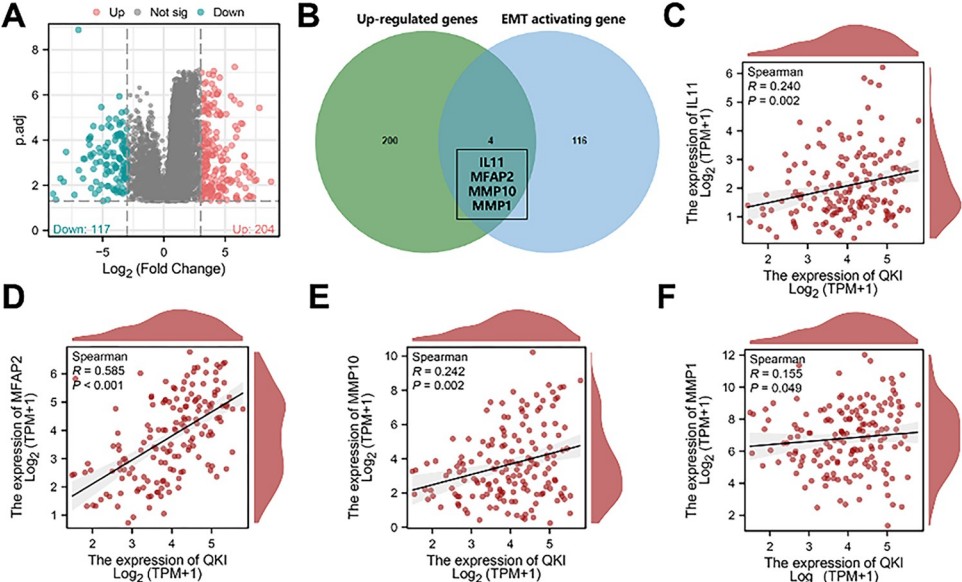

**Fig 3. Correlation analysis of QKI and EMT-related genes.** Volcano map of TCGA-ESCA dataset differential expression analysis. Red dots represent significantly upregulated genes, blue dots represent significantly downregulated genes, and black dots represent genes with no difference in expression (A). Venn diagram of the intersection of 204 significantly upregulated genes in oesophageal cancer and 120 EMT-activated genes (B). The expression correlation analysis results of QKI and IL-11, MFAP2, MMP10 and MMP1 are presented (C-F).

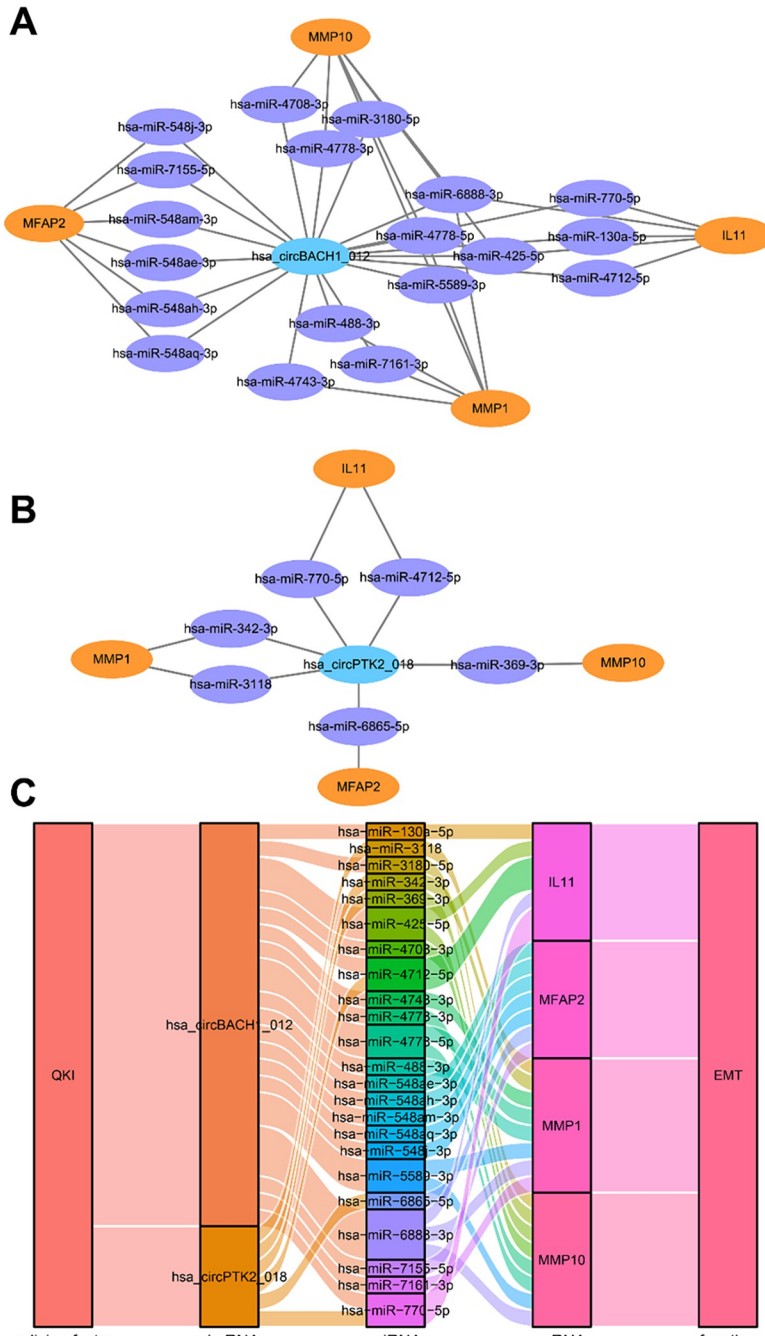

**Fig 4. Correlation analysis between EMT-related genes and survival of oesophageal cancer patients and QKI expression.** The hsa_circ_0006646 and hsa_circ_0061395 downstream miRNA-mRNA regulatory networks (A-B). The mechanism by which QKI promotes the EMT process by regulating the circRNA-miRNA-mRNA regulatory network (C).

biomarkers of oesophageal cancer, and hsa_circ_0001946 has potential as a prognostic bio-marker [29]. Studies have shown that hsa_circ_0006948 [30], circRAD23B [31], hsa_-circ_0030018 [32] and circRNA_100876 [33] can participate in the formation of EMT in oesophageal cancer cells, thus promoting the proliferation, migration and invasion of

**Table 2. The targets of has_circ_0006646.**

| circBank ID | Circbase ID | length | miRNA ID (miR_ID) | miRNAda binding | Targetscan binding |
|---|---|---|---|---|---|
| hsa_circPTK2_018 | hsa_circ_0006646 | 394 | has-miR-3118 | 234 | 251 293 258 299 |
| | | | has-miR-342-3p | 334 | 349 355 |
| | | | has-miR-369-3p | 181 | 193 199 |
| | | | has-miR-4712-5p | 84 | 100 106 |
| | | | has-miR-6865-5p | 280 | 181 186 |
| | | | has-miR-770-5p | 86 | 100 106 |

oesophageal cancer cells. Therefore, identifying the possible molecular mechanism by which EMT is inhibited in oesophageal cancer cells can delay the malignant progression of oesophageal cancer.

In this study, the expression dataset of oesophageal cancer-related genes was obtained from the GEO, TCGA, GTEx and MSigDB databases. Through the integration of data information, it was found that the variable shear factor QKI was closely related to the epithelial mesenchymal transformation (EMT) of oesophageal cancer, and QKI might promote the EMT process by activating the expression of IL-11, MFAP2, MMP10 and MMP1. At the same time, based on the TCGA SpliceSeq database and the GEO dataset GSE189830, 50 coding genes related to QKI and 72 genes encoding differentially expressed circRNAs were identified, and the intersection of the two genes was obtained, namely, BACH1 (hsa_circ_0006646) and PTK2 (hsa_circ_0061395). Therefore, it is speculated that the variable shear factor QKI can promote the generation of hsa_circ_0006646 and hsa_circ_0061395. RNA binding proteins play a crucial role in the biogenesis and degradation of circRNAs. QKI has been proven to participate in the development of cancer by regulating circRNAs [34]. For example, a study showed that QKI can promote the biogenesis of circSLC7A6, which is a type of circRNA with tumour inhibitory activity [35]. In addition, some studies have shown that the RBP CPSF6 can participate in the

**Table 3. The targets of has_circ_0061395.**

| circBank ID | Circbase ID | length | miRNA ID (miR_ID) | miRNAda binding | Targetscan binding |
|---|---|---|---|---|---|
| hsa_circBACH1_012 | hsa_circ_0061395 | 1542 | has-miR-4743-3p | 931 | 1053 1466 1512 446 805 1059 1471 1517 452 811 |
| | | | has-miR-488-3p | 185 | 1121 132 235 1126 137 240 |
| | | | has-miR-130a-5p | 5851312 | 1328 599 1334 605 |
| | | | has-miR-4778-3p | 1441 | 1451 1454 1458 1460 |
| | | | has-miR-4778-5p | 1498 | 1466 1512 1472 1518 |
| | | | has-miR-548ae-3p | 13985 | 152 98 158 105 |
| | | | has-miR-548ah-3p | 86138 | 152 98 158 105 |
| | | | has-miR-548am-3p | 86138 | 152 98 158 105 |
| | | | has-miR-548j-3p | 86 | 152 98 158 105 |
| | | | has-miR-3180-5p | 268 | 285 292 |
| | | | has-miR-425-5p | 1238 | 1253 1259 |
| | | | has-miR-4712-5p | 470 | 485 491 |
| | | | has-miR-5589-3p | 378 | 393 399 |
| | | | has-miR-6888-3p | 1265 | 475 480 |
| | | | has-miR-7155-5p | 1213 | 1224 1230 |
| | | | has-miR-7161-3p | 556 | 1046 1051 |
| | | | has-miR-770-5p | 468 | 485 491 |
| | | | has-miR-4708-3p | 401 | 1237 421 997 1242 427 1002 |
| | | | has-miR-548aq-3p | 86138 | 152 98 158 105 |

**Table 4. Binding of miRNA and mRNA.**

| miRNA | gene | Position in the UTR | Seed match | Context++score | Context++score percentile |
|---|---|---|---|---|---|
| has-miR-425-5p | IL11 | 435–442 | 8mer | -0.46 | 99 |
| has-miR-770-5p | | 62–68 | 7mer-m8 | -0.24 | 93 |
| has-miR-4712-5p | | 62–68 | 7mer-m8 | -0.24 | 93 |
| has-miR-6888-3p | | 412–418 | 7mer-m8 | -0.23 | 94 |
| has-miR-130a-5p | | 393–399 | 7mer-m8 | -0.21 | 92 |
| has-miR-6865-5p | MFAP2 | 174–180 | 7mer-m8 | -0.25 | 93 |
| has-miR-7155-5p | | 329–335 | 7mer-m8 | -0.25 | 96 |
| has-miR-548am-3p | | 406–413 | 8mer | -0.21 | 99 |
| has-miR-548aq-3p | | 406–413 | 8mer | -0.21 | 99 |
| has-miR-548ae-3p | | 406–413 | 8mer | -0.21 | 99 |
| has-miR-548j-3p | | 406–413 | 8mer | -0.21 | 99 |
| has-miR-548ah-3p | | 406–413 | 8mer | -0.21 | 99 |
| has-miR-7155-5p | | 1764–1771 | 8mer | -0.36 | 98 |
| has-miR-7155-5p | | 1455–1461 | 7mer-1A | -0.19 | 92 |
| has-miR-7155-5p | | 1886–1892 | 7mer-1A | -0.18 | 91 |
| has-miR-5589-3p | MMP10 | 208–215 | 8mer | -0.59 | 99 |
| has-miR-4708-3p | | 119–126 | 8mer | -0.56 | 99 |
| has-miR-4778-3p | | 108–115 | 8mer | -0.40 | 98 |
| has-miR-6888-3p | | 15–21 | 7mer-m8 | -0.25 | 95 |
| has-miR-369-3p | | 62–68 | 7mer-m8 | -0.23 | 98 |
| has-miR-3180-5p | | 155–161 | 7mer-m8 | -0.21 | 95 |
| has-miR-4778-5p | | 190–196 | 7mer-m8 | -0.21 | 96 |
| has-miR-425-5p | | 138–144 | 7mer-m8 | -0.22 | 95 |
| has-miR-6888-3p | MMP1 | 331–338 | 8mer | -0.41 | 98 |
| has-miR-5589-3p | | 388–395 | 8mer | -0.40 | 98 |
| has-miR-488-3p | | 324–331 | 8mer | -0.36 | 99 |
| has-miR-4778-5p | | 108–115 | 8mer | -0.33 | 98 |
| has-miR-7161-3p | | 258–264 | 7mer-m8 | -0.29 | 97 |
| has-miR-4743-3p | | 332–339 | 8mer | -0.28 | 98 |
| has-miR-342-3p | | 30–36 | 7mer-m8 | -0.24 | 96 |
| has-miR-3118 | | 394–400 | 7mer-m8 | -0.24 | 91 |

pathogenesis of oesophageal cancer by regulating circCPSF6 [36]. CircRUNX1 mediated by insulin-like growth factor 2 mRNA binding protein 2 can promote the progression of ESCC by regulating the miR-449b-5p/FOXP3 axis, and circRUNX1 has the potential to become a diagnostic marker and therapeutic target for ESCC patients [37]. Through bioinformatics analysis, we found that QKI might promote hsa_circ_0006646 and has_circ_0061395 generation. Circ_0061395 can promote the development of hepatocellular carcinoma by regulating the miR-1182/SPOCK1 signalling axis [38]. Prompt circ_0061395 can play a role as a cancer-promoting circRNA in cancer. Therefore, the role of QKI regulates circ_0061395 in oesophageal cancer deserves further study.

Then, we further analysed the molecular mechanism by which QKI promotes EMT in oesophageal cancer by regulating the circRNA-miRNA-mRNA network. We found that hsa_-circ_0006646 may release the targeted inhibition of IL-11, MFAP2, MMP10 and MMP1 by competitively binding 19 miRNAs. Moreover, hsa_circ_0061395 may release the targeted inhibition of IL-11, MFAP2, MMP10 and MMP1 by competitively binding six miRNAs. Some studies have found a circRNA-miRNA-mRNA regulatory mechanism composed of 29

circRNAs, 2 miRNAs and 10 mRNAs, the regulatory network by which circRNAs may participate in the occurrence and development of ESCC was analysed to some extent [39]. Other studies have shown that the hsa_circ_0006948/miR-490-3p/HMGA2 signalling axis can regulate the EMT process in ESCC and promote the progression of oesophageal cancer [30]. Another study found that circRNA_100367 can enhance the radioresistance of oesophageal cancer cells by regulating the miR-217/Wnt3 pathway [40]. QKI may affect the occurrence and development of cervical cancer by selectively splicing and regulating gene expression [41]. Based on the above findings, we speculate that QKI may promote the occurrence and development of oesophageal cancer by regulating 19 miRNAs downstream of hsa_circ_0006646 and 6 miRNAs downstream of hsa_circ_0061395 to relieve the targeted inhibition of EMT-related genes (IL11, MFAP2, MMP10, MMP1). There are few studies on the detailed regulatory mechanism of QKI and circRNA in cancer. However, some studies have shown that the expression of QKI increased during EMT and that it could be used as a miR-200 target to inhibit EMT and tumorigenesis [24]. Therefore, the molecular mechanism of QKI and circRNAs in cancer needs further study in the future.

In summary, this study shows that the variable shear factor QKI regulates two key circRNAs, hsa_circ_0006646 and hsa_circ_0061395, and downstream related miRNAs can relieve the targeted inhibition of EMT-related genes (IL11, MFAP2, MMP10, MMP1) and promote the occurrence and development of oesophageal cancer, providing a new theoretical basis for screening prognostic markers of oesophageal cancer patients.

## Supporting information

**S1 File.**
(ZIP)

## Author Contributions

**Data curation:** Yishuang Cui, Yanan Wu, Yingze Zhu.

**Methodology:** Lanxiang Huang, Ziqian Hong.

**Software:** Wei Liu, Mengshi Zhang.

**Writing – original draft:** Yishuang Cui.

**Writing – review & editing:** Yishuang Cui, Xuan Zheng, Guogui Sun.

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
