## [Decision Letter · Decision Letter 0]

29 Mar 2023

PONE-D-23-05659The possible molecular mechanism underlying the involvement of the variable shear factor QKI in the epithelial-mesenchymal transformation of oesophageal cancerPLOS ONE

Dear Dr. sun,

Thank you for submitting your manuscript to PLOS ONE. After careful consideration, we feel that it has merit but does not fully meet PLOS ONE’s publication criteria as it currently stands. Therefore, we invite you to submit a revised version of the manuscript that addresses the points raised during the review process.

We look forward to receiving your revised manuscript.

Kind regards,

Matthew Cserhati, Ph.D

Academic Editor

PLOS ONE

Journal Requirements:

   "General Project of National Natural Science Foundation of China (82172658); Tangshan small cell lung cancer medical-industrial integration precision diagnosis and treatment basic innovation team (21130203D)"  

Additional Editor Comments:

Dear Dr. Sun,

We have reviewed your paper and we think that it needs major revisions. i myself have also reviewed the paper and have these comments to make, see below.

Thanks, Matthew Cserhati

Review of manuscript PONE-D-23-05659

a. The abstract conclusion is too short, only one sentence.

b. “In 2020, there will be 604000 new cases of oesophageal cancer and more than 540000 deaths”, should be 604,000 and 540,000.

c. P. 11: briefly describe what FPKM stands for. Also what is ASE?

d. P 11: “Since these data were obtained from public databases, no ethical approval or informed consent was needed.” This should go into some sort of ethics declaration at the end of the paper.

e. P. 11: describe what a shear diagram is briefly.

f. P. 12: how did you combine the TCGA and GTEx datasets (Results, 1st paragraph)?

g. “upregulated compared with that in normal control samples” should be compared with those in …

h. P. 13: Isn’t 0.2 > |r| too low of a threshold?

i. P. 17: put the targets of has_circ_0006646 into a table, and refer to this table in the text (suggestion).

Reviewers' comments:

Reviewer's Responses to Questions

**Comments to the Author**

1. Is the manuscript technically sound, and do the data support the conclusions?

Reviewer #1: Yes

Reviewer #2: Partly

2. Has the statistical analysis been performed appropriately and rigorously? 

Reviewer #1: Yes

Reviewer #2: I Don't Know

3. Have the authors made all data underlying the findings in their manuscript fully available?

Reviewer #1: Yes

Reviewer #2: No

4. Is the manuscript presented in an intelligible fashion and written in standard English?

Reviewer #1: Yes

Reviewer #2: No

5. Review Comments to the Author

Reviewer #1: The manuscript “The possible molecular mechanism underlying the involvement of the variable shear factor QKI in the epithelial-mesenchymal transformation of oesophageal cancer” demonstrated variable shear factor QKI promotes hsa_circ_0006646 and hsa_circ_0061395 generation, which ultimately promotes the EMT process in oesophageal cancer by regulating the circRNA-miRNA-mRNA network.

Comment:

1. The tenses should to be corrected on line 1 of P9.

2. The sentence should describe more accurately on line 3 of P9.

3. The period should be in English format on line 2 of P11.

4. Figure 1A has no legend; Figure 2A should provide a clear image; Figure 2D, 3B, 4A-4B should be more standardized; Each image in Figure 1-4 should provide a clear border, and you can refer to other papers in this journal.

5. The table format should refer to other papers in this journal.

6. References should be in a uniform format, for example, reference 10 does not write the volume issue of the journal; reference 29 writes the month and day.

Reviewer #2: This paper presents a study to disclose that molecular mechanism of the variable shear factor QKI in epithelial mesenchymal transformation of Oesophageal cancer using the GEO, TCGA and GTEX studies. Authors considered different analyses including differential expression analysis (normal vs tumor tissues), gene set enrichment analysis and network analyses using Cytoscape software. However, there are questions that limit my enthusiasm of the paper, as outlined below.

1- Authors mentioned the FPKM data was included in analyses, however for all the plots TPM was mentioned. Is it FPKM or TPM? I assume the logTPM (or FPKM) plus a constant was applied? Please clarify this part and fix all the axis or captions for figures (if needed).

a. The differential expression analyses at Limma package are for counts data or log transformed expression data (FPKM or TPM).

b. If FPKM was considered in the analyses, why not to include TPM (both TCGA and GTEX) since it is a kind gene length and library size normalized data.

c. It is important to share number of features or genes in analyses and any pre-process steps including remove low expressed genes, etc.

2- To increase the number of normal control tissue samples, GTEX was considered in addition to the TCGA. Authors combined these two studies. How about the potential batch(s) between these two cohorts? Any assessment was considered before integration?

3- At the method section, please include more details about all analyses. Authors usually mentioned the R packages or soft wares (e.g., Cytoscape), while it is important to share more details. As an example

a. Differential expression analysis was done by limma package, while it is important to share more details about the Bayesian approach or moderated t statistics (just briefly)

4- FDR or q-value or adjusted p value? Please keep consistency for these terms across manuscript and the figures.

a. Authors mentioned most of the significant results based on p-values (GSEA, DE analysis). Why not adjusted for multiple test?

5- Authors mentioned the median cut-off for expression (Low vs High) at GSEA analysis. I’m wondering to know the binary expression (Low vs High) was applied for differential expression analysis? I couldn’t follow. If yes, please mentioned at DE analysis section.

6- To predict the circRNA-miRNA-mRNA binding relationship, how the cut-off -0.2 was selected?

7- Figure 1: TPM or FPKM? What does star sign represent in Figure 1A? Please add details on caption.

8- The upset Figure 2A is not clear. Please re-generate the figure and add details about 7 groups. The color bar on Heatmap (green to red) represents which statistic? Please clarify.

9- Table 1 only include the p-value. Why not the multiple correction tests approach included?

10- Figure 3A only include TCGA? Or TCGA-GTEX? Is it log10 p-value or adj p-value? Please keep consistency and more precise about the title and axis and caption for figures.

11- For each scatter plot. Please add the line as well.

12- To have reproducible research, is it possible to share code (GitHub) and data?

6. PLOS authors have the option to publish the peer review history of their article (what does this mean?). If published, this will include your full peer review and any attached files.

Reviewer #1: **Yes: **Yufeng Li

Reviewer #2: No

---

## [Author Response · Author response to Decision Letter 0]

23 May 2023

a. The abstract conclusion is too short, only one sentence.

Response: Thank you for your comments from the academic editor. I have revised the original text and marked it in red. The content is as follows:

Variable shear factor QKI promotes hsa_circ_0006646 and hsa_circ_0061395 generation, and downstream related miRNAs can relieve the targeted inhibition of EMT-related genes (IL11, MFAP2, MMP10, MMP1) and promote the occurrence and development of oesophageal cancer, providing a new theoretical basis for screening prognostic markers of oesophageal cancer patients.

b. “In 2020, there will be 604000 new cases of oesophageal cancer and more than 540000 deaths”, should be 604,000 and 540,000.

Response: Thank you for your comments from the academic editor. I have revised the original text and marked it in red.

c. P. 11: briefly describe what FPKM stands for. Also what is ASE?

Response: We are sorry for this mistake. In fact, the data format used in the differential analysis part of this study is TPM not FPKM, and we have provided its full names “transcripts per million (TPM) reads”. Meanwhile, we have unified the descriptions between the figure and methods. In addition, we have already detailed ASE “atomic simulation environment” at its first appearance in the text.

d. P 11: “Since these data were obtained from public databases, no ethical approval or informed consent was needed.” This should go into some sort of ethics declaration at the end of the paper.

Response: Thank you for your comments from the academic editor. I have added an ethical declaration at the end of the paper and marked it in red. The content is as follows:

Ethics Declaration 

Since TCGA、GTEx 、GEO databases were obtained from public databases, no ethical approval or informed consent was needed.

e. P. 11: describe what a shear diagram is briefly.

Response: Thank you for your comments from the academic editor. The content is as follows:

We have revised it to “an UpSet diagram of alternative splicing (schematic diagram of different alternative splicing methods) was drawn”.

f. P. 12: how did you combine the TCGA and GTEx datasets (Results, 1st paragraph)?

Response: Thank you for your comments from the academic editor. In the section of Screening of differentially expressed mRNAs and circRNAs, we have added details “The ComBat R package was used to remove the batch effect of data collected from TCGA and GTEx databases, after which data were merged.”.

g. “upregulated compared with that in normal control samples” should be compared with those in …

Response: Thank you for your comments from the academic editor. I have revised the original text and marked it in red. The content is as follows:

After data correction, an analysis of differential expression showed that the expression of QKI in tumour tissue samples of oesophageal cancer patients was significantly upregulated compared with those in normal control samples.

h. P. 13: Isn’t 0.2 > |r| too low of a threshold?

Response: Thank you for your comments from the academic editor. In the correlation analysis results, we observed that the |r| of most results is mainly distributed within 0.1-0.3, that is, weak correlation. It can also be seen from Table 1 that if the screening criterion was set to |r| > 0.3, only 4 coding genes were screened, which was not conducive to intersecting with another gene set. In general, when multiple screening conditions exist at the same time, we appropriately lowered the screening criteria of |r|, and finally determined |r| > 0.2 as the threshold.

i. P. 17: put the targets of has_circ_0006646 into a table, and refer to this table in the text (suggestion).

Response: Thank you for your comments from the academic editor. The targets of has_circ_0006646 has been shown in Table 3.

Reviewer #1:

1. The tenses should to be corrected on line 1 of P9.

Response: Thank you for your comments from the reviewer. I have revised the original text and marked it in red. The content is as follows:

Based on the GEO, TCGA and GTEx databases, we reveal the possible molecular mechanism of the variable shear factor QKI in epithelial mesenchymal transformation (EMT) of oesophageal cancer.

2. The sentence should describe more accurately on line 3 of P9.

Response: Thank you for your comments from the reviewer. I have revised the original text and marked it in red.

3. The period should be in English format on line 2 of P11.

Response: Thank you for your comments from the reviewer. I have revised the original text and marked it in red.

4. Figure 1A has no legend; Figure 2A should provide a clear image; Figure 2D, 3B, 4A-4B should be more standardized; Each image in Figure 1-4 should provide a clear border, and you can refer to other papers in this journal.

Response: Thank you for your comments from the reviewer. The figure has been modified.

5. The table format should refer to other papers in this journal.

Response: Thank you for your comments from the reviewer. The table has been modified.

6. References should be in a uniform format, for example, reference 10 does not write the volume issue of the journal; reference 29 writes the month and day.

Response: Thank you for your comments from the reviewer. I have revised the original text and marked it in red.

Reviewer #2:

1- Authors mentioned the FPKM data was included in analyses, however for all the plots TPM was mentioned. Is it FPKM or TPM? I assume the logTPM (or FPKM) plus a constant was applied? Please clarify this part and fix all the axis or captions for figures (if needed).

Response: We are sorry for this mistake. In fact, the data format used in the differential analysis part of this study is TPM, and we have unified the descriptions between the figure and methods. 

As you said, the data were first converted to log 2(TPM+1) and then subjected to differential analysis, which is also a common method used for analysis of the TPM data. We also referred to the GEPIA2 database (http://gepia2.cancer-pku.cn/#analysis) and adopt this way of drawing.

a. The differential expression analyses at Limma package are for counts data or log transformed expression data (FPKM or TPM).

Response: In this study, the data were first converted to log 2(TPM+1) and then subjected to differential analysis.

b. If FPKM was considered in the analyses, why not to include TPM (both TCGA and GTEX) since it is a kind gene length and library size normalized data.

Response: In fact, the data format used in the differential analysis part of this study is TPM, and we have unified the descriptions between the figure and methods.

c. It is important to share number of features or genes in analyses and any pre-process steps including remove low expressed genes, etc.

Response: Thank you for your suggestion. We have added relevant details in the methods section “The ComBat R package was used to remove the batch effect of data collected from TCGA and GTEx databases, after which data were merged. Then the limma R package was applied to standardize the gene expression data, and analyse the differential expression of QKI in normal control tissue samples and tumour tissue samples from esophageal carcinoma patients. The non-parametric Wilcoxon rank sum test was used when normality assumption was not satisfied…. we selected the path with the most significant difference with the normalized enrichment score (NES) p value and false discovery rate (FDR) q value as the criteria and drew the enrichment plot. Larger |NES| values of each gene set indicate that the gene set exhibits a stronger enrichment signal between the high and low expression groups of QKI. If |NES| values are more than 0, the gene set exhibits higher activity in the high expression group of QKI; if less than 0, the gene set is more active in the low expression group of QKI.”.

2- To increase the number of normal control tissue samples, GTEX was considered in addition to the TCGA. Authors combined these two studies. How about the potential batch(s) between these two cohorts? Any assessment was considered before integration?

Response: Thank you for your suggestion. It is necessary to merge TCGA and GTEx after removing batch effects, and our data analysis also includes this step. But the method description may be too concise, resulting in some details being lost. We have improved the corresponding method sections and have worked hard to review them again. We have added relevant details in the Screening of differentially expressed mRNAs and circRNAs section “The ComBat R package was used to remove the batch effect of data collected from TCGA and GTEx databases, after which data were merged. Then the limma R package was applied to standardize the gene expression data, and analyse the differential expression of QKI in normal control tissue samples and tumour tissue samples from esophageal carcinoma patients. The non-parametric Wilcoxon rank sum test was used when normality assumption was not satisfied”.

3- At the method section, please include more details about all analyses. Authors usually mentioned the R packages or soft wares (e.g., Cytoscape), while it is important to share more details. As an example

a. Differential expression analysis was done by limma package, while it is important to share more details about the Bayesian approach or moderated t statistics (just briefly)

Response: Thank you for your suggestion. We have improved the corresponding method section and have worked hard to review it again. We have added relevant details in the Screening for differentially expressed mRNAs and circRNAs section “Then the limma R package was applied to standardize the gene expression data, and analyse the differential expression of QKI in normal control tissue samples and tumour tissue samples from esophageal carcinoma patients. The non-parametric Wilcoxon rank sum test was used when normality assumption was not satisfied”.

4- FDR or q-value or adjusted p value? Please keep consistency for these terms across manuscript and the figures.

a. Authors mentioned most of the significant results based on p-values (GSEA, DE analysis). Why not adjusted for multiple test?

Response: We have checked the text and made corresponding revision to keep consistency for these terms across manuscript and the figures. In addition, we only performed correction for differential analysis and GSEA results but not for the results of correlation analysis. 

5- Authors mentioned the median cut-off for expression (Low vs High) at GSEA analysis. I’m wondering to know the binary expression (Low vs High) was applied for differential expression analysis? I couldn’t follow. If yes, please mentioned at DE analysis section.

Response: The specific process of GSEA in this study is as follows: first, samples were divided into a high-expression QKI group and a low-expression QKI group. Then, each gene set was subjected to statistical analysis using the GSEA method (this study refers to the HALLMARK dataset) and an enrichment score (ES) was calculated. ES represents expression difference degree of the gene set in high-expression and low-expression samples; for each gene set, a background distribution is generated by random sampling and repeated calculation, after which p-value and FDR are calculated to judge whether the gene set is significantly enriched in the high-expression and low-expression samples. In the section of Gene set enrichment analysis (GSEA), we have detailed “In this study, we analysed the TCGA-ESCA dataset based on the QKI median expression value to analyse the high and low expression groups of QKI and compared the enriched pathway differences between the two groups by referring to the HALLMARK dataset. Then, we selected the path with the most significant difference with the normalized enrichment score (NES) p value and false discovery rate (FDR) q value as the criteria and drew the enrichment plot. Larger |NES| values of each gene set indicate that the gene set exhibits a stronger enrichment signal between the high and low expression groups of QKI. If |NES| values are more than 0, the gene set shows higher activity in the high expression group of QKI; if less than 0, the gene set is more active in the low expression group of QKI.”. 

6- To predict the circRNA-miRNA-mRNA binding relationship, how the cut-off -0.2 was selected?

Response: In the correlation analysis results, we observed that the |r| of most results is mainly distributed within 0.1-0.3, that is, weak correlation. It can also be seen from Table 1 that if the screening criterion was set to |r| > 0.3, only 4 coding genes were screened, which was not conducive to intersecting with another gene set. In general, when multiple screening conditions exist at the same time, we appropriately lowered the screening criteria of |r|, and finally determined |r| > 0.2 as the threshold.

7- Figure 1: TPM or FPKM? What does star sign represent in Figure 1A? Please add details on caption.

Response: In Figure 1, it is TPM. In addition, we have explained “* p.adj < 0.05” in the legend.

8- The upset Figure 2A is not clear. Please re-generate the figure and add details about 7 groups. The color bar on Heatmap (green to red) represents which statistic? Please clarify.

Response: We have modified this figure and have provided detailed information in the results text “As shown in Fig. 2A, there was a large amount of alternative splicing in esophageal carcinoma. The height of the columns in the figure indicates the number of genes that undergo a certain alternative splicing. The first three columns are the individual parts of exon skip (ES), alternate terminator (AT) and alternate promoter (AP), the fourth column indicates the common part of ES and AP, and the fifth column indicates the common part of ES, AT and AP, which fully showed that the proportion of ES, AT and AP in esophageal carcinoma was relatively high.”. In addition, in this figure legend, we have detailed “Heatmap for differentially expressed genes in the GSE189830 dataset (normal group, n = 3; tumour group, n = 3) (C). The expression values are represented in different colors; red indicates high expression, and green indicates low expression. Darker node colors indicate more extreme high or low expression of the respective gene.”.

9- Table 1 only include the p-value. Why not the multiple correction tests approach included?

Response: We only performed correction for differential analysis but not for the results of correlation analysis (Gao Z,Xu J,Zhang Z, et al. A Comprehensive Analysis of METTL1 to Immunity and Stemness in Pan-Cancer. Front Immunol. 2022;13:795240. doi:10.3389/fimmu.2022.795240; Zhan X,Cheng J,Huang Z, et al. Correlation Analysis of Histopathology and Proteogenomics Data for Breast Cancer. Mol Cell Proteomics. 2019;18 (8 suppl 1):S37-S51. doi:10.1074/mcp.RA118.001232). We guess it may be because in the data analysis based on databases, for example, TCGA, the data were standardized. may be because in the process of data analysis based on databases such as TCGA, the data will be standardized and analyzed with log2(value+1). We chose Spearman or Pearson for correlation analysis based on the results of data normality test, so the correction of p value may not be so important. According to your suggestion, we have checked relevant literature and found that the p-value can be corrected by Bonferroni correction or FDR in correlation analysis, and we will also pay attention to it in future research.

10- Figure 3A only include TCGA? Or TCGA-GTEX? Is it log10 p-value or adj p-value? Please keep consistency and more precise about the title and axis and caption for figures.

Response: In fact, Figure 3A includes TCGA and GTEx. We have re-plotted this figure. The abscissa in the figure represents logFC, and the ordinate represents the corrected p value. We have changed the label of the ordinate to “p.adj”. In addition, we have marked the points of up-regulation and down-regulation in the figure, as well as the specific number for better understanding.

11- For each scatter plot. Please add the line as well.

Response: Thank you for your suggestion. We have added the line for each scatter plot.

12- To have reproducible research, is it possible to share code (GitHub) and data?

Response: We are more than willing to share data and if necessary, we can provide all raw data, upload as an attachment, or upload to GitHub.

---

## [Decision Letter · Decision Letter 1]

6 Jun 2023

PONE-D-23-05659R1The possible molecular mechanism underlying the involvement of the variable shear factor QKI in the epithelial-mesenchymal transformation of oesophageal cancerPLOS ONE

Dear Dr. sun,

Thank you for submitting your manuscript to PLOS ONE. After careful consideration, we feel that it has merit but does not fully meet PLOS ONE’s publication criteria as it currently stands. Therefore, we invite you to submit a revised version of the manuscript that addresses the points raised during the review process.

We look forward to receiving your revised manuscript.

Kind regards,

Abdul Rauf Shakoori

Academic Editor

PLOS ONE

Journal Requirements:

Reviewers' comments:

Reviewer's Responses to Questions

**Comments to the Author**

1. If the authors have adequately addressed your comments raised in a previous round of review and you feel that this manuscript is now acceptable for publication, you may indicate that here to bypass the “Comments to the Author” section, enter your conflict of interest statement in the “Confidential to Editor” section, and submit your "Accept" recommendation.

Reviewer #1: (No Response)

Reviewer #2: All comments have been addressed

2. Is the manuscript technically sound, and do the data support the conclusions?

Reviewer #1: (No Response)

Reviewer #2: Yes

3. Has the statistical analysis been performed appropriately and rigorously? 

Reviewer #1: (No Response)

Reviewer #2: I Don't Know

4. Have the authors made all data underlying the findings in their manuscript fully available?

Reviewer #1: (No Response)

Reviewer #2: Yes

5. Is the manuscript presented in an intelligible fashion and written in standard English?

Reviewer #1: (No Response)

Reviewer #2: Yes

6. Review Comments to the Author

Reviewer #1: The manuscript “The possible molecular mechanism underlying the involvement of the variable shear factor QKI in the epithelial-mesenchymal transformation of oesophageal cancer” demonstrated the variable shear factor QKI regulates two key circRNAs, hsa_circ_0006646 and hsa_circ_0061395, and downstream related miRNAs could relieve the targeted inhibition of EMT-related genes (IL11, MFAP2, MMP10, MMP1) and promoted the occurrence and development of oesophageal cancer, providing a new theoretical basis for screening prognostic markers of oesophageal cancer patients.

Comment:

1.The tenses need to be unified, such as line 17 in the Abstract, “promotes” should be “promoted”.

2. Line 1 in the Methods, “database” should be “databases”, pay attention to the use of single and plural numbers.

3. Line 19 in the Methods, Gene set enrichment analysis should be Gene set enrichment analysis (GSEA).

4. Line 20 in the Methods, “an” should be deleted.

5. Line 34 in the Methods, epithelial mesenchymal transformation should be EMT, abbreviation should be used.

6. Line 12 in the Results, “Fig. 1.” in the figure title should be “Figure 1”.

7. Line 21 in the Results, full name of ES, AP and AT should be given when first appeared.

8. Line 38 in the Results, “was” should be “is”.

9. Line 44 in the Results, the blank between “3” and “B” should be deleted.

10. Line 56 in the Results, the beginning space should be deleted.

11. Line 71 in the Results, “gastric cancer” should be “oesophageal cancer”.

12. The references should be checked carefully according to the PUBMED. And the format of references should be uniform.

Reviewer #2: Authors addressed most of the comments, however two important comments as follows haven't seen any appropriate response:

1- For multiple test correlation analyses, no correction to control FDR or FWER. Why postpone to the future work?

2- No GitHub link was shared. While, it is important to share codes to publish research reproducible (specially when the data are public and no limitation to get access)

I'm going to ask Editor to make decision for the above two comments.

Thank you

7. PLOS authors have the option to publish the peer review history of their article (what does this mean?). If published, this will include your full peer review and any attached files.

Reviewer #1: No

Reviewer #2: No

---

## [Author Response · Author response to Decision Letter 1]

19 Jun 2023

Reviewer #1: 

Comment:

1. The tenses need to be unified, such as line 17 in the Abstract, “promotes” should be “promoted”.

Response: Thank you for your comments from the reviewer. I have revised the original text and marked it in red. The content is as follows:

We further identified the significantly upregulated circRNAs and their corresponding coding genes in oesophageal cancer, screened the EMT-related genes that were significantly positively correlated with QKI expression, predicted the circRNA-miRNA binding relationship through the circBank database, predicted the miRNA-mRNA binding relationship through the TargetScan database, and finally obtained the circRNA-miRNA-mRNA network through which QKI promoted the EMT process.

2. Line 1 in the Methods, “database” should be “databases”, pay attention to the use of single and plural numbers.

Response: Thank you for your comments from the reviewer. I have revised the original text and marked it in red. The content is as follows:

The TCGA databases (https://www.cancer.gov) was used to download the gene expression data of normal control and oesophageal cancer patients, and the downloaded data format was transcripts per million (TPM) reads.

3. Line 19 in the Methods, Gene set enrichment analysis should be Gene set enrichment analysis (GSEA).

Response: Thank you for your comments from the reviewer. I have revised the original text and marked it in red. The content is as follows:

Gene set enrichment analysis (GSEA) is a computational method for determining whether a priori-defined gene set shows statistically significant consistent differences between two biological states.

4. Line 20 in the Methods, “an” should be deleted.

Response: Thank you for your comments from the reviewer. I have made modifications in the original text.

5. Line 34 in the Methods, epithelial mesenchymal transformation should be EMT, abbreviation should be used.

Response: Thank you for your comments from the reviewer. I have revised the original text and marked it in red. The content is as follows:

Obtaining EMT-related genes from the MSigDB database

6. Line 12 in the Results, “Fig. 1.” in the figure title should be “Figure 1”.

Response: Thank you for your comments from the reviewer. I have revised the original text and marked it in red. The content is as follows:

Figure 1 Analysis of differential expression of QKI and pathway enrichment.

7. Line 21 in the Results, full name of ES, AP and AT should be given when first appeared.

Response: Thank you for your comments from the reviewer. I have revised the original text and marked it in red. The content is as follows:

The first three columns are the individual parts of exon skip (ES), alternate terminator (AT), and alternate promoter (AP).

8. Line 38 in the Results, “was” should be “is”.

Response: Thank you for your comments from the reviewer. I have revised the original text and marked it in red. The content is as follows:

QKI expression is significantly positively correlated with the expression of EMT-related genes (IL11, MFAP2, MMP10, MMP1)

9. Line 44 in the Results, the blank between “3” and “B” should be deleted.

Response: Thank you for your comments from the reviewer. I have made modifications in the original text.

10. Line 56 in the Results, the beginning space should be deleted.

Response: Thank you for your comments from the reviewer. I have made modifications in the original text.

11. Line 71 in the Results, “gastric cancer” should be “oesophageal cancer”.

Response: Thank you for your comments from the reviewer. I have revised the original text and marked it in red. The content is as follows:

Figure 4 Correlation analysis between EMT-related genes and survival of oesophageal cancer patients and QKI expression.

12. The references should be checked carefully according to the PUBMED. And the format of references should be uniform.

Response: Thank you for your comments from the reviewer. I have revised the original text and marked it in red.

Reviewer #2: Authors addressed most of the comments, however two important comments as follows haven't seen any appropriate response:

1- For multiple test correlation analyses, no correction to control FDR or FWER. Why postpone to the future work?

Response: Thank you for your suggestion. We also strongly agree with the necessity of multiple tests. Therefore, we have added the following code to the correlation analysis. However, we have found that after multiple corrections, the correlation analysis results have not changed. The screenshot of the correlation analysis results before and after correction is shown below. The correlation analysis results can be found in the attached document, which was revised back together. The results before correction were "corResult before. txt", and the results after correction were "corResult after. txt". We have also modified table1 by replacing the original p value with the corresponding result of p. adjusted. Finally, we considered the reason why the correlation analysis results before and after correction did not change. Comparing the p-values before and after correction, the difference between p value and p. adjusted is relatively small, and we also use the r value as the screening criterion, indicating that the r value may have a greater impact on the screening results. Thank you again for your suggestion. We will also pay more attention to the rigor of bioinformatics analysis.

2- No GitHub link was shared. While, it is important to share codes to publish research reproducible (specially when the data are public and no limitation to get access).

Response: Thank you for your suggestion. My original data GitHub link is https://github.com/06cys2023/plos-one.

---

## [Editor Report · Decision Letter 2]

26 Jun 2023

The possible molecular mechanism underlying the involvement of the variable shear factor QKI in the epithelial-mesenchymal transformation of oesophageal cancer

PONE-D-23-05659R2

Dear Dr. sun,

We’re pleased to inform you that your manuscript has been judged scientifically suitable for publication and will be formally accepted for publication once it meets all outstanding technical requirements.

Kind regards,

Abdul Rauf Shakoori

Academic Editor

PLOS ONE
---

## [Editor Report · Acceptance letter]

29 Jun 2023

PONE-D-23-05659R2 

The possible molecular mechanism underlying the involvement of the variable shear factor QKI in the epithelial-mesenchymal transformation of oesophageal cancer 

Dear Dr. Sun:

I'm pleased to inform you that your manuscript has been deemed suitable for publication in PLOS ONE. Congratulations! Your manuscript is now with our production department. 

Kind regards, 

on behalf of

Dr. Abdul Rauf Shakoori 

Academic Editor

PLOS ONE